# The Rheological Performance and Structure of Wheat/Acorn Composite Dough and the Quality and In Vitro Digestibility of Its Noodles

**DOI:** 10.3390/foods10112727

**Published:** 2021-11-07

**Authors:** Qian Zhang, Jiangtao Yu, Kui Li, Junqiing Bai, Xiuyun Zhang, Yifan Lu, Xiangxiang Sun, Wenhao Li

**Affiliations:** 1College of Food Science and Engineering, Northwest A&F University, Xianyang 712100, China; z-grace@nwafu.edu.cn (Q.Z.); 1011204319@nwafu.edu.cn (X.Z.); Luyifan0222@163.com (Y.L.); sunxiangxiang@nwafu.edu.cn (X.S.); 2Yangling Hesheng Irradiation Technologies Co., Ltd., Xianyang 712100, China; likui@ssn-hs.com (K.L.); baijunqing@ssn-hs.com (J.B.)

**Keywords:** noodles, acorn flour, dough rheology, texture, nutrition quality

## Abstract

Wheat flour was partially replaced by debittered acorn flour (DAF) with 0%, 10%, 15%, 20% as well as 25%. Rheological properties of wheat/acorn dough and quality and in vitro digestibility of its noodles were determined. Results showed that DAF addition significantly improved pasting viscosity and dough stability time while excessive addition weakened the protein network and decreased maximum fermentation height. Furthermore, noodles with substitutions exhibited promising technological properties as a food ingredient for noodle making (higher hardness, chewiness, gumminess, firmness, and less cooking time) but poor extensibility, smaller lightness values, and a slight deterioration in cooking quality. Furthermore, PCA and correlation analysis demonstrated a significant relationship between textural and cooking properties and pasting and mixing parameters. Moreover, SEM images of acorn noodles presented coarser surfaces but a tighter cross-section structure. Finally, in vitro digestibility results indicated that DAF addition significantly reduced the susceptibility of the starches to enzyme hydrolysis, while the addition of acorn flour slightly decreased the overall acceptability. Thus, the partial substitution of wheat flour with acorn flour can favorably be used in noodles formulation.

## 1. Introduction

Oak (*Quercus* spp.) is one of the most species-rich tree groups that comprise more than 500 species plus several hybrids distributed throughout temperate and tropical regions [1]. Nut fruits produced by oaks, i.e., acorns, have been extensively studied and confirmed the great biotechnological potential for pharmaceutics, biomaterials, biofuel production as well as food processing [2,3,4]. Acorns contain various nutritional components such as carbohydrates, proteins, lipids, fibers, vitamins, minerals, and different sterols [5]. In addition, Strong interest has been focused on its diverse biological activities, including antioxidant and anti-inflammatory activities that come from the numerous bioactive compounds such as saponin, phenolic compounds, and terpenoids [1,6]. There is no doubt that acorns could be considered functional food or an alternative food ingredient offering health benefits.

With a rapidly growing population (approximately 9 billion in 2050), the global food system must make reasonable improvements to adapt to the concept of sustainable development [7]. Exploring wild edible plants as substitutes for other agricultural products or as new ingredients in the food industry appears to be a viable option [8]. However, acorns are far from insufficient for food development and utilization [1,9]. Thus, it requires additional research to develop new acorn products that promote human health and have tremendous market value.

The addition of functional flour obtained from acorns in bread, cake, and biscuit have been recently evoked [5,10]. However, Acorn is rich in tannin, which generates the inedible astringent taste and behaves as an anti-nutrients, and appropriate methods are required to remove it [11]. Furthermore, acorn flour contains no gluten proteins, which confer viscoelasticity to the dough, while wheat proteins pose unique viscoelastic characteristics during dough preparation and are extensively used in food products [12]. Therefore, combining the good processing performance of wheat flour with the nutrition function of acorn flour, the development of acorn staple foods, and the formation of new product types are beneficial to the development of both acorn and flour industries.

Although acorn flour in making gluten-free foods has been extensively studied, there are no studies conducted over combining acorn flour with wheat flour to produce Chinese noodles. In this context, the work aimed to assess acorn flour’s making performance, in vitro digestibility, and sensory properties as a partial substitution for wheat flour in noodle making and explore the relationship between dough characteristics and noodle properties. Hopefully, the results would provide insights for developing functional acorn staple food industry with high profitability and thus promote global sustainability.

## 2. Materials and Methods

### 2.1. Materials

Commercial wheat flour (WF) was obtained from Shaanxi Bull Flour Co., Ltd., BaoJi, China. Acorn fruits were randomly collected from acorn planting areas located in Luonan County, Shaanxi province. Other chemicals and reagents used in the current study were of analytical grade and purchased from Sanli Chemical Reagent Co, Yangling, China.

The premixed WF and debittered acorn flour (DAF) after removing tannins with substituting 10%, 15%, 20%, and 25%, the premixed WF and debittered acorn flour (DAF) were thoroughly homogenized by rotating and shaking in a self-sealing bag and named DAF10, DAF15, DAF20, DAF25, respectively.

### 2.2. Tannin Removal Treatment

The removal of tannin was performed as described by Zhang et al. [11] with slight changes. The Acorn powder was first passed through a 40-mesh to remove large impurities. Acorn powder was soaked with distilled water at the ratio of 1: 30 [w (g)/v (mL)] for 48 h in an oven (DHG-9123A model, Shanghai Jinghong Instrument Co., Ltd., Shanghai, China) at 45 °C with the water changed every 8 h. The precipitation was collected and dried at 40 °C for 24 h; the sample was crushed and passed through the 100-mesh sieve twice to obtain DAF.

### 2.3. Proximate Analysis

Proximate compositions of moisture, protein, fat, and ash of samples were analyzed using AACC methods (AACC 2000). A starch assay kit determined the total starch contents (Megazyme International Ireland Ltd., Bray, Ireland). The dough was washed and gluten retained was collected and weighed to determinate the content wet gluten. Apparent amylose content was determined according to the method of Chrastil [13].

### 2.4. Rheological Tests of Mixed Dough

#### 2.4.1. Pasting Properties Analysis

Pasting properties of WF, DAF, and mixed flour were studied with a Rapid Visco Analyzer (RVA, RVA-4500, Perten Instruments, Stockholm, Sweden) using the method by Bhattacharya, et al. [14]. Flour (3 g, 14%mb) was weighted and dispersed in distilled water. The flour slurry (28 g total weight) was equilibrated at 50 °C for 1 min, then heated to 95 °C at a rate of 12 °C/min and kept for 2.5 min before cooling to 50 °C at the same rate, and holding for 2 min.

#### 2.4.2. Mixing Characterization

Thermomechanical properties of dough were measured by Mixolab2 (Chopin Technologies, Villeneuve-la-Garenne, France) using the “Chopin +” protocol described by Moreira et al. [15]. Samples were weighted according to the software calculation corresponding to the predicted water absorption and moisture content (f.b., corrected to 14% moisture basis) and placed into the Mixolab bowel with distilled water to reach a total weight of 75 g dough. The target torque was 1.1 ± 0.5 N. The dough was firstly developed at 30 °C for 8 min and followed by the raised temperature to 90 °C at a rate of 4 °C/min and held for 7 min, and then the temperature was reduced to 50 °C at a rate of 4 °C/min and kept for 5 min. The total analysis time is 45 min. The parameters were obtained and calculated as described by the previous reports [15].

#### 2.4.3. Rheofermentometer Rheological Measurements

Dough fermentation properties were determined with a Rheofermentometer F4 (Chopin technologies, Villeneuve-la-Garenne, France) following the method of Huang et al. [16]. Reconciled dough (315 g) was put into a bucket and fermented for 3 h at 30 °C. The maximum dough weight, Hm, and the total volume of gas, VT was determined and recorded.

### 2.5. Preparation of Acorn Noodle

After pre-experiment, it was determined that the final replacement ratio of acorn powder is 0%, 10%, 15%, 20%, and 25%. Chinese noodle fabrication with different proportions of acorns was as follows: 200 g mixed flour (addition of acorn flour at 0%, 10%, 15%, 20%, 25%), 40% distilled water, and 0.75% salt were put into a small dough mixer (KVC30, Delong Electric Co., Ltd., Shanghai, China), and mixed for 10 min. After incubated for 40 min at room temperature, the dough crumbles were passed through a small noodle machine 6 times. The final size of the noodles was 1 mm thick, 3.0 mm wide, and 220 mm long.

### 2.6. Noodle Quality Evaluation

#### 2.6.1. Cooking Properties of Acorn Noodles

Optimum cooking time

The optimum cooking time was determined according to the method of Guo et al. [17]. Generally, noodles (10 g) were cooked in 300 mL boiling water. Samples were taken every 15 s for 2 min, and the noodle core was pressed between two transparent glasses to observe when the white core disappeared, which is the optimum cooking time.

Water absorption and cooking loss

Water absorption and cooking loss were analyzed as described by Raungrusmee et al. [18] with some modifications. Noodle sample (W1) was cooked for the optimum cooking time in 300 mL boiling water. The cooked noodles were picked out and weigh as W2 after removing the surface moisture with filter paper. The boiled water was evaporated in an oven at 105 °C overnight to a constant weight to determine the weight of the remaining solid matter and record it as W3. The acorn noodles’ water absorption and cooking loss were calculated using Equations (1) and (2), respectively.
(1)$$\mathrm{Water}\;\mathrm{absorption}\;\left(\%\right)=\frac{{(W}_{2}-W_{1})}{W_{1}}\times 100$$
(2)$$\mathrm{Cooking}\;\mathrm{loss}\;\left(\%\right)=\frac{W_{3}}{W_{1}}\times 100\;$$
where W_1_ is the weight of fresh acorn noodles, W_2_ is the weight of cooked acorn noodles, and W_3_ is the weight of remaining solid matter.

#### 2.6.2. Color Measurement

Color attributes for samples were measured using a CR-310 Chromameter (Minolta Corporation, Tokyo, Japan) and expressed as L* (lightness), a* (+redness/−greenness) and b* (+yellowness/−blueness) values.

#### 2.6.3. Texture Properties of Acorn Noodles

TPA, shear, and tensile tests of fresh and cooked acorn noodles were performed using a TA-XT Plus texture analyzer (Stable Microsystems Ltd., Godalming, UK) and refer to the method of Liu et al. [19]. The measurements of cooked noodles were carried out within 15 min after cooking. The instrument was calibrated using a 1 kg load cell. P/6, A/LKD, and A/SPR probes were used for TPA, shear, and tensile analysis, respectively. The parameter settings were as follows. For TPA tests: pre-test speed, 1 mm/s; post-test speed, 2 mm/s; strain, 75%. For shear analysis: pre-test speed, 1 mm/s; post-test speed, 2 mm/s; strain, 95%. For tensile tests: distance, 50 mm; pre-test speed, 1 mm/s; test speed, 2 mm/s; post-speed, 10 mm/s; trigger force, 5 g.

#### 2.6.4. Microscopy Observing

Fresh and cooked noodles were dried at −80 °C overnight and lyophilized the next day, then the samples were fixed on the sample stage with conductive double-sided tape and sputtered with a thin layer of gold. The surface- and cross-sectional images were observed 500× at an accelerating voltage of 5 kV by SEM (Nova Nano SEM 450, FEI, Hillsboro, OR, USA).

#### 2.6.5. In Vitro Digestibility

The multi-enzymatic method was carried out to analyze noodle digestibility as Bustos et al. [20] described with slight modification. Triplicate samples (50 mg) of cooked noodles were dispersed in 10 mL phosphate-buffered saline (PBS, pH 6.9, 0.12 M NaCl, 2.7 mM KCl and 0.01 M phosphate buffer salts). The slurries were then acidified to a pH of 1.5 with 1 M HCl, and 0.1 mL pepsin solution (115 U/mL) was added, followed by incubation at 37 °C for 30 min. After that, samples were neutralized by adjustment (1 M NaOH) to the pH of 6.9, then 0.5 mL porcine pancreatic α-amylase solution (110 U/mL) was added to it, and volume was made up to 25 mL using PBS (pH 6.9). Each tube was incubated for 3 h at 37 °C. Aliquots (1 mL) were withdrawn every 30 min and boiled at 100 °C for 5 min to inactivate the enzyme. After cooling, 3 mL 0.4 M acetate buffer (pH 4.75) was added, followed by 20 μL glucoamylase, after which the tubes were incubated for 45 min at 55 °C. The hydrolyzed glucose was quantified using the GOPOD reagent. The glucose content was converted into hydrolyzed starch by multiplying a factor of 0.9. The kinetics of starch hydrolysis was fitted to the first-order equation:(3)$${C=C}_{\infty }(1-e^{-\mathrm{kt}})$$
where C refers to the percentage of starch hydrolyzed at time t; C_∞_ corresponds to the equilibrium percentage of starch hydrolyzed after 180 min; k is the first-order rate coefficient.

#### 2.6.6. Sensory Evaluation

20 food professionals performed sensory evaluation following 9 point hedonic scale for sensory attributes viz., appearance, color, flavor, softness, thickness, elasticity, overall acceptability.

### 2.7. Statistical Analysis

All assays were performed thrice, and the values are expressed as mean ± standard deviation (SD). Data were subjected to analysis of variance (ANOVA) followed by Turkey’s test (*p* ≤ 0.05) using Minitab Statistical Software (Minitab v18.1, Minitab Inc., State College, PA, USA). Correlation analysis and PCA (principal component analysis) were conducted using origin 2021.

## 3. Results

### 3.1. Physicochemical Properties of Flour

The chemical composition of WF and DAF was summarized in Table 1. The moisture content of WF (14.21%) was significantly higher than that of DAF (11.46%). Furthermore, the apparent amylose content of WF and DAF was 18.98% and 20.00%, respectively. Other basic chemical compositions of WF was fat (0.70%), protein (11.46%), total starch (67.35%), and ash (0.35%). Wet gluten content of wheat dough was 30.79%. Compared to WF, DAF contained less protein (3.81%), and ash (0.40%); but more fat (1.07%) and starch (77.97%). This is consistent with the results in the previous literature suggesting that acorn flour contains more than 55% of starch, 2.75–8.44% protein, and 0.7–7.4% fat [21].

### 3.2. Rheological Properties of Mixed Dough

#### 3.2.1. Pasting Properties

Pasting profiles are shown in Figure 1a and parameters including peak viscosity (PV), trough viscosity (TV), final viscosity (FV), peak time (PT), and gelatinization time (GT) are presented in Table 2. It can be observed the partial substitution of wheat flour for acorn flour contributed to a significant increase in PV, TV, FV, setback (SB), and PT, while a significant decrease in the breakdown (BD). PV is the maximum viscosity during the heating process and indicates the binding ability of starch [22]. Mixed flour with greater PV suggested a more efficient ability to hold water, resulting from starch being more accessible to water under conditions of lower protein content. A similar negative correlation between protein content and PV was also observed by Marcoa and Rosell [23]. Furthermore, TV is minimum viscosity value during the gelatinization period while BD is the difference between PV and TV. TV and BD are used as indicators predicting the hot paste stability under shear [24]. TV significantly increased while BD decreased with increasing substitution level, revealing enhanced heating and shear stress resistance.

Final viscosity is associated with the ability of starch to form a gel structure after cooling and setback is the difference between FV and TV. FV and SB are influenced by protein, starch, amylose, amylopectin, and amylose/amylopectin ratio [14]. The exhibition of higher FV and SB values of mixed flour may be attributed to the higher amylose content and lower protein content. Higher amylose contents caused an increased in leached amylose and lower protein content increased the accessibility between leached amylose and, ultimately, paving a way for more affinity towards retrogradation. This result was well corroborated with the previous literature [24] in the inversely relationship between FV and protein content. Peak time is the time required to reach the peak viscosity. Interestingly, the higher peak time of mixed flour than wheat flour counteracts the finding of Ajo [25] where a higher peak time was associated with higher protein content. However, Ma and Baik [26] found that PT is related to both protein content and quality. In this study, a possible explanation is that factors such as type of starch, lipid, fiber, and polyphenols also influenced the peak time.

#### 3.2.2. Mixing Properties

Mixing characteristics of dough with different proportions of acorn flour are shown in Figure 1b and Table 3. DAF20 and DAF25 exhibited higher water absorption (W.A) values than other samples, indicating more moisture required to ensure the full hydration of the ingredients. Water absorption was significantly affected by damaged starch, total protein and pentosan content [27,28]. This result may be attributed to that addition of acorn flour increased the insoluble fiber content in the dough and potentially destroyed original starch and protein structure of wheat flour, resulting in more hydrophilic groups combined with water molecules, thereby increasing the water absorption. Dough development time (D.T.T) is the time needed to reach C1. It reflects the strength of gluten, While C2 is the minimum torque of dough consistency measuring protein weakening as a function of temperature and mechanical work [15]. Adding acorn flour significantly decreased the D.T.T and C2 values, revealing that acorn flour intensively weakened the protein network and thus the elasticity of the dough. Therefore, it can be assumed that excessive acorn flour replacing wheat flour will finally reduce cooking characteristics product properties. However, on the other hand, a proper substitution ratio could improve the time for making noodles while ensuring product quality from a technical point of view. 

Stability time (Stab.) is the time for the maximum consistency of the dough to remain constant during mixing. Stab. increased with increasing substitution level, suggesting that acorn flour improved the strength of the dough and the tolerance for mixing, which might be attributed to the covalent and non-covalent interactions between proteins and polyphenols derived from acorn flour. The results align with previous reports where the substitution of wheat flour by acorn flour could increase stab. [29,30]. During the heating process, the maximum and minimum torque were measured and defined as C3 and C4, respectively. Substitutions showed a higher viscosity (larger C3 value) and significantly better thermal stability (larger C4 value) compared with the non-treated group, which was consistent with previous work [31]. Additionally, the thermal weakening also exhibited the same tendency. 

C5 and the difference between C5 and C4 values have been closely associated with the retrogradation tendency of the flour. However, no statistically significant difference was observed between the different proportions of substitutions. Meanwhile, there was no significant difference between different substitution levels (0%, 10%, 15%, 20%, 25%) in mechanical weakening. α is used to measure the rate of weakening of the protein network, while β represents the gelatinization speed. The increasing replacement rate of acorn flour resulted in a rise in α value and showed a significant difference when the addition ratio raised to 20%. Samples with adding substitutions presented significantly higher β values than wheat flour, but there is no significant difference between different replacement ratios. These results showed that partial replacement of wheat flour with acorn flour could speed up the denaturation speed of the protein network and the gelatinization speed of starch.

#### 3.2.3. Fermentation Properties

The effect of adding acorn flour on the fermentation properties of dough is shown in Table 2. Hm, the maximum dough height is an indirect index evaluating the performance of yeast and the overall microstructure of the studied dough. Adding acorn flour significantly reduced the Hm from 14.2 of control to 4.1 of 20% replacement ratio, suggesting weakened strength expanding and holding CO_2_ gas cells and destroyed dough network, which is consistent with the results of the mixing assay. VT is the total volume of gas released during fermentation. It was represented that acorn flour had no significant effect on CO_2_ production.

### 3.3. Noodle Quality

#### 3.3.1. Cooking Quality

As shown in Table 4, acorn flour had a noticeable impact on the cooking quality parameters of the noodles. The optimum cooking time significantly decreased from 240 s to 210 s as the substitution ratio increased, which may be due to the diluted protein content inhibiting the protein network formation. The results are in agreement with those obtained by Xie et al. [32]. Water absorption and cooking loss are crucial indexes to assess the cooking quality of noodles [33]. The addition of acorn flour significantly decreased the water absorption compared with the control noodle, which corroborates the view that the longer the optimum cooking time, the greater the water absorption rate [34]. Furthermore, the cooking loss of substitution values was significantly higher than the control, representing more solid content leaking and the turbidity of noodle soup.

Correlation analysis was performed among rheological properties of dough, and noodle quality showed in Figure 2. There was a highly significant negative relevance between water absorption and pasting parameters except for BD, Stab., β, and T.W. In contrast, a significant positive correlation was found between water absorption and α. In contrast, the cooking loss was significantly positively correlated with the pasting parameters while negatively with α, indicating that RVA and Mixolab could predict acorn noodles’ quality.

#### 3.3.2. Color Attributes

Three color parameters, L*, a*, and b* of different substitutions levels, were evaluated and summarized in Table 4. The lightness (L*) significantly decreased from 80.22 of the control to a minimum value of 59.8 at DAF20, while the yellowness (b*) and redness (a*) of mixed flour were significantly higher than that of the control and reached the maximum when the ratio was 10% and 15%, respectively. L* was correlated significantly and inversely with a* and b* (Figure 3). These observed differences might be associated with the original color of acorn flour and the presence of natural pigments.

#### 3.3.3. Textural Properties of Fresh and Cooked Noodles

Textural parameters could often be taken as an essential indicator that affects the taste and noodle acceptance. The effect of acorn flour substitution (0%, 10% 15%, 20%, 25%) on textural properties, including TPA, shear, and stretch parameters of fresh and cooked noodles, are listed in Table 5. For TPA, the hardness and gumminess of raw noodles significantly increased with increased substitution levels until a peak at 20%, after which the hardness plateaued and the gumminess decreased. However, no significant difference in resilience and cohesiveness was found between the treatment and control fresh noodles.

Furthermore, the adhesiveness of fresh noodles didn’t show substantial differences until the acorn concentration reached 25%. Fresh noodles’ chewiness and springiness were first reduced and subsequently increased, and the DAF10 showed minimal springiness and chewiness. Moreover, acorn flour significantly affected cooked noodles’ hardness, cohesiveness, gumminess, and chewiness. Correlation analysis showed that the hardness of fresh noodles was highly positively correlated with the most pasting properties, W.A., Stab., T.W, α, and β (Figure 2, while negatively with BD. These results supported the opinion of Luo et al. [35] that the strength of the gluten work largely governs hardness. At the same time, chewiness and springiness are highly associated with the overall gluten network.

Adding acorn flour significantly enhanced the firmness, work of shear (WF), and resistance to extension (RE) but reduced the extensibility of fresh noodles (Table 5). Furthermore, the analyses showed statistically strong negative correlations among extensibility, hardness, and firmness (Figure 2). After cooking, the firmness, WF, decreased while RE increased. The firmness and WS of cooked substitutions were significantly higher than the control, while the RE was the opposite. Furthermore, there was no significance in extensibility between cooked substitutions and the control. These discrepancies between the fresh and cooked noodles might be due to the different available water content and lost and deformed protein during cooking.

#### 3.3.4. Scanning Electron Microscopy (SEM)

Figure 3 shows the SEM images of the surface and cross-section of the noodles with different substitution levels. The surface of fresh noodles exhibited a continuous matrix with starch granules entrapped in the gluten protein network (Figure 3A1–E1). It revealed that noodle samples presented coarser surfaces even along with pores as the substitution level increased. Similar findings were also observed by Goñi et al. [36], which may be explained by the rapid formation of a strong gluten network, resulting in fewer surface connections and increased cooking losses. In addition, boiled noodles showed compact and smooth surfaces because of the swelling and gelatinization of starch, and it is hard to distinguish between starch and protein (Figure 3A2–E2).

When comparing the cross-section images (Figure 3A3–E3), it can be observed that the dough structure became more compact with the increased adding acorn flour, indicating that protein-starch interactions were enhanced. However, when the substitution ratio increased to 20%, holes and cracks appeared in the dough structure. Furthermore, in the cross-sectional images of cooked noodles, the fracture of the network structure inside the dough was more visible (Figure 3A4–E4). This phenomenon may be attributed to the fact that adding acorn flour relatively reduced the gluten content in the dough.

#### 3.3.5. In Vitro Digestibility

All the experimental values were substituted into the equation C = C_∞_ (1 − e ^− kt^), and k (kinetic constant) and C_∞_ (Equilibrium value of hydrolysis percentage) were obtained in Table 6. The R2 values of the fitted curves’ R2 values range from 0.965 to 0.990 (Table 6), suggesting the model had significant goodness of fit statistics. Of note, C_∞_ of the control was significantly higher than acorn-enriched noodles, while no significance between different substitution levels. Moreover, no significance was detectable in the k value among the whole samples.

Hydrolysis Index (HI) and Predicted Glycemic Index (PGI) were calculated as previously described [37] and shown in Table 6. The substitutions exhibited significantly lower HI values than the control, and PGI followed the same trend, which phenolics may explain in acorn flour. Similar results were reported by Soong et al. [37]. However, it is worth paying attention to an interesting phenomenon: C_∞_, HI, and GI all reached the minimum value when the substitution ratio was 15% and then slightly increased. It should be recognized that the enzymatic susceptibility of starch depends not only on the noodles components but also on the structural state of the substrate. Excessive substitution destroyed the protein starch matrix, making the starch more susceptible to degradation by enzymes, and this result coincides with other findings [38,39].

#### 3.3.6. Sensory Evaluation

Sensory ratings of cooked noodles from appearance, color, flavor, softness, stickiness, elasticity, and overall liking were evaluated by a trained panel and tabulated in Table 7. The scores of softness, stickiness, and elasticity of substitutions were slightly lower than those of the control group but no significant difference was found, which could be due to the significantly higher hardness, firmness value and lower extensibility. These results were comparable with the TPA analysis of cooked noodles. In addition, panelists reported a special flavor of noodles containing acorn flour, which may explain the lower flavor scores of acorn noodles. However, this flavor is not unacceptable according to their moderate scores. Furthermore, appearance and color were negatively influenced by the application of acorn flour due to its darker color, which may be accountable for the lower overall liking scores of the substitutions than the control. Among all substitutions, the noodle with 10% acorn flour received the highest overall liking scores and scores of it were found statistically (*p* < 0.05) similar to the control. Nevertheless, all noodles prepared with acorn flour were liked moderately and liked slightly by panelists. 

### 3.4. PCA

Principal component analysis (PCA) was carried out to visualize the similarities and differences between mixed flours and the interrelationship between the properties of flours, doughs, and noodles, and the results are shown in Figure 4. The variance contribution of PC1 and PC2 were 74.6% and 17.9%, respectively, accounting for 92.5% of the information of the original data set cumulatively. PCA scores (Figure 4a) reported the apparent separation, indicating that flour quality changed dramatically by the addition. The distance between any two samples on the score plot is proportional to the degree of difference between their properties. Furthermore, the viscosity, most mixing parameters, overall liking, firmness, gumminess, adhesiveness, and hardness were significant factors contributing to PC1 (Figure 4b). All these properties were correlated with each other, while M.W, springiness, and chewiness dominated PC2. 

TF, WS, firmness, gumminess, adhesiveness, and hardness are close to DAF20 and DAF25, indicating that they are highly correlated. Furthermore, the pasting properties, mixing properties, and textural properties are close in the plot, suggesting that they are correlated, consistent with the correlation analysis. Furthermore, DAF10 and DAF15 are not far away from each other, indicating a similarity in their nature. In addition, the L*, HI, and PGI were located close to WF while opposite to a*, b*, and cooking loss, which confirmed a positive effect of acorn flour on glycemia.

## 4. Discussion

Partial replacement of wheat flour by debittered acorn flour significantly affected the rheological properties of dough and the quality of noodles. The pasting, mixing, textural properties, and in vitro digestibility were improved by adding acorn flour. In addition, the dough added acorn powder showed longer stabilization time, better gelatinization stability, and a more ordered cross-sectional structure. However, high levels of wheat flour replacement caused a deterioration in cooking quality and an unattractive appearance. Furthermore, the Correlation and PCA analysis indicated that dough’s rheological properties determine the nature of the noodles to a certain extent. Given the above results, this experiment recommends 10% to 15% acorn flour replacing wheat flour to produce acorn noodles from the nutrition value quality perspective. However, further study is needed to investigate the applicability of acorn flour in other food and the mechanism by which the acorn flour influences the characteristics of pasta.

## Figures and Tables

**Figure 1 foods-10-02727-f001:**
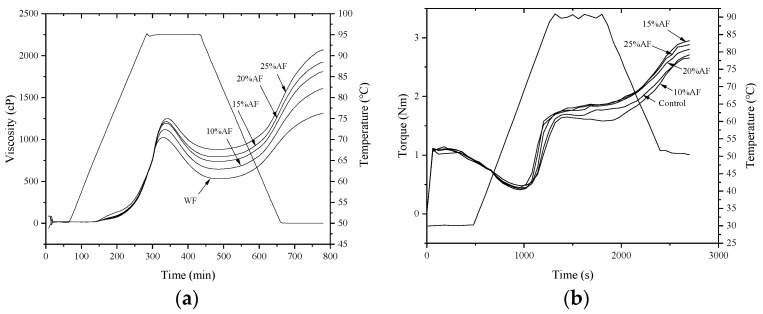
Pasting (**a**) and mixing (**b**) profiles of mixed flour with different substitution levels.

**Figure 2 foods-10-02727-f002:**
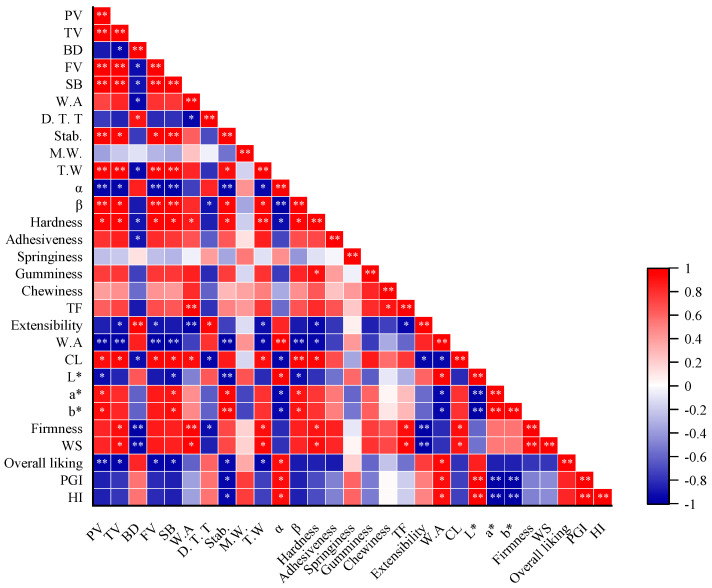
Correlation analysis of cooking properties and textural properties of acorn noodles and pasting properties and Mixolab characteristics of the acorn-wheat flour mixture. TF tensile force, W.A water absorption, CL cooking loss, PV peak viscosity, TV trough viscosity, BD breakdown, FV final viscosity, SB setback, D.T.T dough development time, Stab. stability, M.W mechanical weakening, T.W thermal weakening, * *p* < 0.05, ** *p* < 0.01.

**Figure 3 foods-10-02727-f003:**
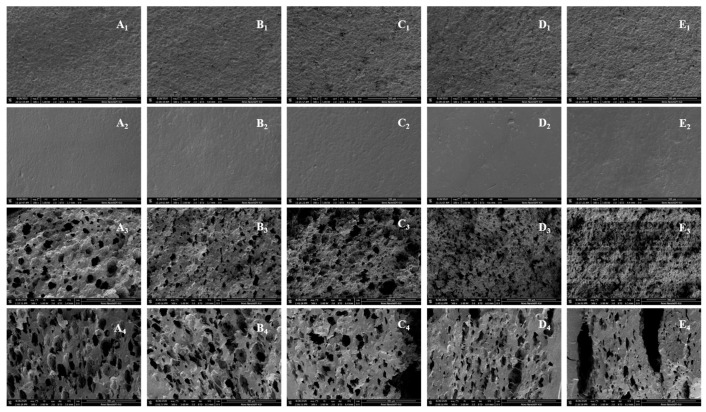
Scanning electron micrographs (SEM) (×500) of the surface and cross-section of noodles with different proportions of acorn flour ((**A**) 0%; (**B**) 10%; (**C**) 15%; (**D**) 20%; (**E**) 25%; (**1**) the surface of raw noodles; (**2**) the surface of cooked noodles; (**3**) Cross-section of raw noodles; (**4**) Cross-section of cooked noodles.

**Figure 4 foods-10-02727-f004:**
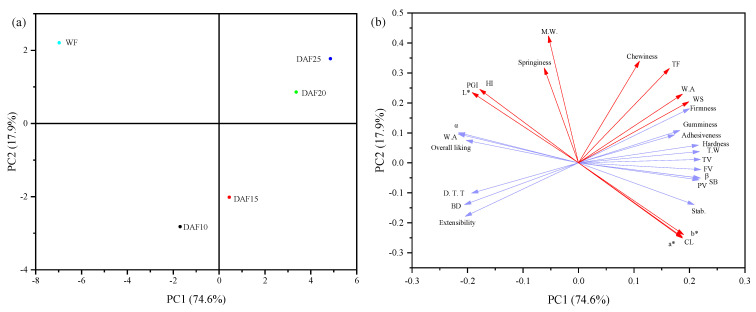
Score plot (**a**) and loading plot (**b**) of the first component (PC1) and the second component (PC2) describing the relationship of various flour, dough, and noodles characteristics. Samples nomenclature used is described above.

**Table 1 foods-10-02727-t001:** Basic physical and chemical composition of acorn and wheat flour ^a,b,c^.

Sample	Moisture (%)	Fat (%)	Protein (%)	Amylose (%)	Total Starch (%)	Ash (%)	Wet Gluten (%)
WF	14.21 ± 0.01 ^a^	1.07 ± 0.37 ^a^	12.46 ± 0.47 ^a^	18.98 ± 0.17 ^b^	67.35 ± 0.64 ^b^	0.35 ± 0.03 ^a^	30.79 ± 0.20 ^a^
DAF	11.46 ± 0.01 ^b^	1.12 ± 0.16 ^a^	3.81 ± 0.05 ^b^	20.00 ± 0.31 ^a^	77.97 ± 0.78 ^a^	0.14 ± 0.03 ^b^	ND

^a^ Data are represented by mean of triple ± standard deviation and different letters within the same column indicate a significant difference (*p* < 0.05). ^b^ The number after DAF refers to the substitution level (g/100 g of wheat flour). ^c^ ND means not detected.

**Table 2 foods-10-02727-t002:** Pasting and fermentation parameters of wheat flour with different substitution levels ^a,b,c^.

Samples	Pasting Properties							Fermentation Properties	
	PV (cP)	TV (cp)	BD (cP)	FV (cP)	SB (cP)	PT (min)	GT (°C)	H_m_/mm	V_T_/mL
WF	1023 ± 52 ^b^	531 ± 17 ^d^	492.0 ± 35.4 ^a^	1310 ± 39 ^d^	780 ± 22 ^d^	5.47 ± 0.00 ^b^	86.85 ± 0.57 ^a^	14.2 ± 2.0 ^a^	1488 ± 132 ^a^
DAF10	1119 ± 37 ^ab^	646 ± 23 ^c^	472.5 ± 13.4 ^a^	1606 ± 54 ^c^	959 ± 31 ^c^	5.57 ± 0.05 ^ab^	86.50 ± 0.00 ^a^	10.9 ± 1.5 ^ab^	1508 ± 9 ^a^
DAF15	1196 ± 57 ^a^	738 ± 20 ^bc^	458.5 ± 36.1 ^ab^	1808 ± 42 ^b^	1070 ± 22 ^b^	5.64 ± 0.05 ^ab^	86.45 ± 0.14 ^a^	4.2 ± 0.6 ^c^	1506 ± 10 ^a^
DAF20	1213 ± 47 ^a^	792 ± 22 ^ab^	420.5 ± 24.8 ^ab^	1921 ± 54 ^ab^	1128 ± 32 ^ab^	5.64 ± 0.05 ^ab^	86.43 ± 0.04 ^a^	4.1 ± 0.5 ^c^	1464 ± 221 ^a^
DAF25	1251 ± 18 ^a^	880 ± 12 ^a^	371.0 ± 5.7 ^bc^	2067 ± 21 ^a^	1186 ± 9 ^a^	5.70 ± 0.04 ^a^	86.83 ± 0.53 ^a^	4.8 ± 5.2 ^bc^	1253 ± 84 ^ab^

^a^ PV: peak viscosity; TV: trough viscosity; BD: breakdown; FV: final viscosity; SB: setback; PT: peak time; GT: gelatinization temperature; H_m_: the maximum dough height; V_T_: total volume of gas. ^b^ Data are represented by mean of triple ± standard deviation and different letters within the same column indicate a significant difference (*p* < 0.05). ^c^ The number after DAF refers to the substitution level (g/100 g of wheat flour).

**Table 3 foods-10-02727-t003:** Mixolab parameters of wheat flour with different substitution levels of acorn flour (C1:1.10 ± 0.05 Nm) ^a,b,c^.

Sample	WA (%)	DTT (min)	Stab. (min)	C2 (N·m)	C3 (N·m)	C4 (N·m)	C5 (N·m)	C5 − C4 (N·m)	α	β
Control	60.0	2.76 ± 0.05 ^a^	5.1 ± 0.0 ^c^	0.47 ± 0.00 ^a^	1.65 ± 0.01 ^b^	1.58 ± 0.02 ^c^	2.61 ± 0.03 ^a^	1.03 ± 0.04 ^a^	0.046 ± 0.001 ^a^	0.14 ± 0.00 ^b^
DAF10	60.0	1.90 ± 0.81 ^ab^	6.0 ± 0.1 ^b^	0.44 ± 0.01 ^ab^	1.66 ± 0.03 ^ab^	1.75 ± 0.03 ^b^	2.66 ± 0.02 ^a^	0.92 ± 0.04 ^a^	0.052 ± 0.001 ^ab^	0.17 ± 0.01 ^a^
DAF15	60.0	2.55 ± 0.00 ^a^	6.5 ± 0.1 ^ab^	0.44 ± 0.00 ^ab^	1.72 ± 0.01 ^ab^	1.83 ± 0.05 ^ab^	2.82 ± 0.18 ^a^	0.98 ± 0.13 ^a^	0.053 ± 0.000 ^ab^	0.18 ± 0.00 ^a^
DAF20	60.7	0.96 ± 0.02 ^b^	6.6 ± 0.0 ^a^	0.40 ± 0.03 ^b^	1.71 ± 0.05 ^ab^	1.83 ± 0.02 ^ab^	2.76 ± 0.06 ^a^	0.94 ± 0.04 ^a^	0.055 ± 0.001 ^b^	0.20 ± 0.00 ^a^
DAF25	60.8	0.79 ± 0.04 ^b^	6.6 ± 0.2 ^a^	0.41 ± 0.00 ^b^	1.75 ± 0.00 ^a^	1.87 ± 0.02 ^a^	2.87 ± 0.02 ^a^	1.00 ± 0.04 ^a^	0.056 ± 0.000 ^c^	0.20 ± 0.01 ^a^

^a^ W.A.: water absorption; D.T.T.: dough development time. ^b^ Data are represented by mean of triple ± standard deviation and different letters within the same column indicate a significant difference (*p* < 0.05). ^c^ The number after DAF refers to the substitution level (g/100 g of wheat flour).

**Table 4 foods-10-02727-t004:** Cooking quality and color attributes of wheat noodles with different substitution levels ^a,b^.

Sample	Cooking Quality			Color Attributes		
	Cooking Time (s)	Water Absorption (%)	Cooking Loss (%)	Lightness (L*)	Redness (a*)	Yellowness (b*)
Control	240	65.95 ± 3.17 ^a^	1.63 ± 0.06 ^c^	80.22 ± 0.17 ^a^	0.84 ± 0.03 ^c^	15.66 ± 0.09 ^b^
DAF10	225	59.75 ± 0.80 ^ab^	2.54 ± 0.02 ^b^	62.62 ± 0.51 ^b^	5.21 ± 0.03 ^b^	19.28 ± 0.35 ^a^
DAF15	225	57.88 ± 0.02 ^b^	2.48 ± 0.11 ^b^	60.93 ± 0.20 ^c^	5.65 ± 0.03 ^a^	20.03 ± 0.07 ^a^
DAF20	225	56.21 ± 0.90 ^b^	3.61 ± 0.15 ^a^	59.80 ± 0.31 ^cd^	5.75 ± 0.05 ^a^	20.10 ± 0.28 ^a^
DAF25	210	54.90 ± 0.98 ^b^	3.68 ± 0.27 ^a^	60.19 ± 0.46 ^d^	5.63 ± 0.10 ^a^	19.81 ± 0.58 ^a^

^a^ Data are represented by mean of triple ± standard deviation and different letters within the same column indicate significant differences among groups (*p* < 0.05). ^b^ The number after DAF refers to the substitution level (g/100 g of wheat flour).

**Table 5 foods-10-02727-t005:** Textural parameters of fresh wheat noodles with different substitution levels of acorn flour ^a,b,c^.

Sample		TPA							Shear		Stretch	
		Hardness (g)	Adhesiveness (g·s)	Springiness	Cohesiveness	Gumminess	Chewiness	Resilience	Firmness (g)	WS (g·s)	RE (g)	Extensibility (mm)
Fresh	Control	1598 ± 53 ^d^	−5.18 ± 1.08 ^a^	0.54 ± 0.01 ^b^	0.35 ± 0.02 ^c^	525.7 ± 22.7 ^d^	333.0 ± 34.5 ^b^	0.33 ± 0.01 ^ab^	1852 ± 82 ^c^	1033 ± 53 ^d^	21.72 ± 0.76 ^d^	12.15 ± 0.15 ^a^
	DAF10	1752 ± 19 ^c^	−4.85 ± 0.54 ^a^	0.43 ± 0.03 ^c^	0.35 ± 0.03 ^c^	553.9 ± 29.8 ^cd^	228.4 ± 27.9 ^c^	0.30 ± 0.02 ^b^	1895 ± 16 ^c^	1014 ± 51 ^d^	19.40 ± 0.88 ^e^	11.68 ± 0.36 ^a^
	DAF15	1968 ± 66 ^b^	−3.88 ± 0.34 ^a^	0.51 ± 0.06 ^b^	0.34 ± 0.02 ^c^	616.1 ± 49.0 ^c^	281.1 ± 34.0 ^bc^	0.31 ± 0.03 ^ab^	1940 ± 53 ^c^	1134 ± 45 ^c^	21.04 ± 0.45 ^de^	10.9 ± 0.22 ^b^
	DAF20	2241 ± 109 ^a^	−4.43 ± 0.31 ^a^	0.49 ± 0.02 ^bc^	0.37 ± 0.01 ^c^	836.5 ± 52.0 ^a^	440.6 ± 79.5 ^a^	0.34 ± 0.02 ^a^	2149 ± 67 ^b^	1323 ± 61 ^b^	25.48 ± 0.53 ^c^	7.63 ± 0.37 ^c^
	DAF25	2218 ± 57 ^a^	−1.92 ± 0.24 ^b^	0.49 ± 0.05 ^b,c^	0.33 ± 0.01 ^c^	702.2 ± 12.2 ^b^	367.6 ± 59.2 ^ab^	0.30 ± 0.01 ^b^	2328 ± 24 ^a^	1458 ± 47 ^a^	26.32 ± 1.87 ^c^	6.51 ± 0.20 ^d^
Cooked	Control	372 ± 7 ^f^	−13.03 ± 1.02 ^cd^	1.00 ± 0.01 ^a^	0.42 ± 0.01 ^b^	157.0 ± 2.5 ^f^	147.3 ± 5.7 ^d^	0.17 ± 0.00 ^d^	484 ± 18 ^f^	425 ± 16 ^d^	50.70 ± 4.77 ^a^	11.81 ± 1.54 ^a^
	DAF10	444 ± 7 ^e^	−12.67 ± 2.23 ^cd^	0.99 ± 0.00 ^a^	0.44 ± 0.00 ^a^	199.1 ± 2.9 ^e^	192.4 ± 9.2 ^c^	0.16 ± 0.01 ^d^	720 ± 35 ^d^	674 ± 23 ^a^	49.02 ± 1.47 ^a^	12.15 ± 0.91 ^a^
	DAF15	442 ± 21 ^e^	−9.24 ± 0.94 ^c^	0.99 ± 0.04 ^a^	0.45 ± 0.01 ^a^	206.4 ± 10.6 ^e^	217.5 ± 11.5 ^c^	0.17 ± 0.00 ^d^	672 ± 48 ^d^	661 ± 55 ^ab^	47.70 ± 2.24 ^a,b^	10.95 ± 1.15 ^a^
	DAF20	444 ± 20 ^e^	−13.86 ± 3.06 ^d^	1.04 ± 0.22 ^a^	0.46 ± 0.01 ^a^	207.0 ± 8.0 ^e^	225.8 ± 43.2 ^c^	0.17 ± 0.01 ^d^	681 ± 39 ^d^	603 ± 44 ^b^	47.77 ± 2.56 ^ab^	10.69 ± 0.83 ^a^
	DAF25	464 ± 24 ^e^	−11.20 ± 3.07 ^cd^	1.00 ± 0.00 ^a^	0.46 ± 0.01 ^a^	211.2 ± 8.2 ^e^	215.2 ± 24.3 ^c^	0.18 ± 0.01 ^c^	594 ± 45 ^e^	531 ± 25 ^c^	42.75 ± 1.31 ^b^	10.94 ± 0.86 ^a^

^a^ WS and RE indicate work of shear and resistance to extension, respectively. ^b^ Data are represented by mean of triple ± standard deviation and different letters within a column indicate significant differences among groups (*p* < 0.05). ^c^ The number after DAF refers to the substitution level (g/100 g of wheat flour).

**Table 6 foods-10-02727-t006:** Equilibrium Concentration (C_∞_), Kinetic constant (K), Hydrolysis Index (HI), and Predicted Glycemic Index (PGI) for acorn noodles with different substitution levels ^a,b^.

Samples	C_∞_ (%)	k	R^2^	HI (%)	PGI
Control	82.75 ± 0.15 ^a^	0.022 ± 0.002 ^a^	0.970	58.48 ± 1.48 ^a^	71.82 ± 0.81 ^a^
DAF10	67.71 ± 1.55 ^b^	0.022 ± 0.001 ^a^	0.973	47.75 ± 0.24 ^b^	65.92 ± 0.13 ^b^
DAF15	64.88 ± 12.18 ^b^	0.023 ± 0.003 ^a^	0.972	46.63 ± 6.86 ^b^	65.31 ± 3.77 ^b^
DAF20	68.23 ± 1.49 ^a,b^	0.021 ± 0.002 ^a^	0.965	47.64 ± 0.78 ^b^	65.86 ± 0.43 ^b^
DAF25	67.16 ± 2.65 ^b^	0.022 ± 0.005 ^a^	0.990	47.51 ± 1.69 ^b^	65.79 ± 0.93 ^b^

^a^ Data are represented by mean of triple ± standard deviation and different letters within the same column indicate significant differences among groups (*p* < 0.05). ^b^ The number after DAF refers to the substitution level (g/100 g of wheat flour).

**Table 7 foods-10-02727-t007:** Sensory characteristics of fresh wheat noodles with different substitution levels ^a,b^.

Sample	Appearance	Color	Flavor	Softness	Stickiness	Elasticity	Overall Liking
Control	8.18 ± 0.60 ^a^	8.09 ± 0.83 ^a^	7.45 ± 0.82 ^a^	7.64 ± 1.21 ^a^	7.73 ± 1.10 ^a^	7.36 ± 1.03 ^a^	8.18 ± 0.75 ^a^
DAF10	7.09 ± 1.38 ^ab^	7.27 ± 0.79 ^ab^	7.36 ± 0.92 ^a^	6.91 ± 1.04 ^a^	6.82 ± 0.98 ^a^	7.09 ± 1.04 ^a^	7.64 ± 0.81 ^ab^
DAF15	6.09 ± 1.38 ^b^	6.64 ± 0.83 ^b^	6.91 ± 0.94 ^a^	6.55 ± 1.13 ^a^	7.00 ± 0.77 ^a^	6.45 ± 0.69 ^a^	7.00 ± 0.77 ^b^
DAF20	6.82 ± 1.54 ^ab^	6.82 ± 0.78 ^b^	7.00 ± 0.89 ^a^	6.82 ± 1.40 ^a^	7.09 ± 1.04 ^a^	6.55 ± 1.13 ^a^	7.27 ± 1.10 ^ab^
DAF25	6.45 ± 1.37 ^b^	6.91 ± 0.51 ^b^	6.73 ± 0.79 ^a^	6.91 ± 0.83 ^a^	7.00 ± 0.89 ^a^	6.36 ± 1.03 ^a^	6.73 ± 1.10 ^b^

^a^ Data are represented by mean of triple ± standard deviation and different letters within the same column indicate significant differences among groups (*p* < 0.05). ^b^ The number after DAF refers to the substitution level (g/100 g of wheat flour).

## Data Availability

The data presented in this study are available on request from the corresponding author.

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
