# Peer review of "The Rheological Performance and Structure of Wheat/Acorn Composite Dough and the Quality and In Vitro Digestibility of Its Noodles"

_foods, 2021, doi:10.3390/foods10112727_

Round 1

Reviewer 1 Report

In this work, the effect of partial replacement of wheat flour by debittered acorn flour on dough and noodles properties was investigated. The experimental procedure was well carried out. The article is enough interesting and quite well written (I suggest reading again the article and doing a little English editing). Some corrections to be done and some suggestions for the authors are reported in the attached file.

Author Response

Response to Reviewer 1 Comments

Point 1: Please rephrase the following sentence: The effect of partial replacement (0%, 10%, 15%, 20%, 25%) of wheat flour (WF) by debittered acorn flour (DAF) on the rheological properties of dough and textural, cooking quality, and in vitro digestibility, sensory parameters of noodles were determined.

Response 1: The sentence has reorganized as “Wheat flour was partially replaced by debittered acorn flour (DAF) with 0%, 10%, 15%, 20% as well as 25%. Rheological properties of wheat/acorn dough and quality and in vitro digestibility of its noodles were determined”.

Point 2: Convert the word “revelation” to “relationship”.

Response 2: The word has been converted.

Point 3: Line 26: groups of what?

Response 3: Group of tree. In addition, I changed this sentence into “Oak (Quercus spp.) is one of the most species-rich tree groups that…” for more fluent.

Point 4: Line 59: Authors should put as section 2 materials and methods and section 3 results according to journal template.

Response 4: The materials and methods has been put as section 2 and the result has been put as section 3.

Point 5: Line 75: Define acronym before using it.

Response 5: The acronym has been defined in the materials and methods part after adjusting it to section 2.

Point 6: Line 93: You can also cite other papers that found the same results as you, like:10.1094/CCHEM-11-16-0274-R, 10.1016/j.jcs.2019.04.007.

Response 6: The paper whose DOI is 10.1094/CCHEM-11-16-0274-R has been cited and the opinion of the paper has been supplemented to support the result of ours.

Point 7: Line 105-106: Explanation for this? Why these two samples have an higher WA? Could it be related to their composition?

Response 7: Possible explanation has been provided considering factors affecting water absorption and related literature has been cited. The final statement is as follows: Water absorption was significantly affected by damaged starch, total protein and pentosan content [27, 28]. This result may be attributed to that addition of acorn flour increased the insoluble fiber content in the dough and potentially destroyed original starch and protein structure of wheat flour, resulting in more hydrophilic groups combined with water molecules, thereby increasing the water absorption.

Point 8: Line 116: Define what is it. You can define this parameter just one time in material section (after moving it to section2).

Response 8: The definition of stability time has been explained in detail as follows: Stability time (Stab.) is the time for the maximum consistency of the dough to remain constant during mixing.

Point 9: Table 3: Adjust table, hard to read

Response 9: The table has been adjusted for better reading.

Point 10: Line 132: Add blank line before going back to the discussion after tables, figures etc..., check this in the whole manuscript please.

Response 10: We have checked the whole manuscript and added blank line before going back to the discussion after tables and figures.

Point 11: Line 302: This should be section 4.

Response 11: Results part has been adjusted as section 4.

Point 12: Line 350: How did you choose this? Did you measure it on flour?

Response 12: 14% wet basis is the most commonly used benchmark in the mixolab “chopin +” protocol. The moisture content of the flour is measured in advance and entered into the software, then the software will automatically calculate the required flour content and the amount of water added.

Reviewer 2 Report

The article entitled The rheological performance and structure of wheat/acorn composite dough and the quality and in vitro digestibility of its noodles is interesting and present a new perspective of using the acorn flour. The authors must improve their explanation and not to write that a parameter is decreasing or increasing, a scientific approach will be reached if the authors find the  connections between the different compounds from the both flours and the resulting behaviour.

Table 1 - add the wet gluten content

Lines 89-95 - the authors just make an observation, they should improve the scientific explanations

Tabel 2there is a parameter without measuring unit.

Sensory properties are too slight explained. Please improve it

Why you used PCA? 5 samples are sufficient for pca?

Author Response

Response to Reviewer 2 Comments

Point 1: Table 1 - add the wet gluten content.

Response 1: The wet gluten content was determined and added in table 1.

Point 2: Lines 89-95 - the authors just make an observation, they should improve the scientific explanations.

Response 2: Based on your comments, we have added an explanation of the change in viscosity parameters. The supplementary content is as follows: Final viscosity is associated with the ability of starch to form a gel structure after cooling and setback is the difference between FV and TV. FV and SB are influenced by protein, starch, amylose, amylopectin, and amylose/amylopectin ratio [14]. The exhibition of higher FV and SB values of mixed flour may be attributed to the higher amylose content and lower protein content. Higher amylose contents caused an increased in leached amylose and lower protein content increased the accessibility between leached amylose and, ultimately, paving a way for more affinity towards retrogradation. This result was well corroborated with the previous literature [24] in the inversely relationship between FV and protein content. Peak time is the time required to reach the peak viscosity. Interestingly, the higher peak time of mixed flour than wheat flour counteracts the finding of Ajo [25] where a higher peak time was associated with higher protein content. However, Ma and Baik [26] found that PT is related to both protein content and quality. In this study, a possible explanation is that factors such as type of starch, lipid, fiber, and polyphenols also influenced the peak time.

Point 3: Tabel 2 there is a parameter without measuring unit.

Response 3: The unit has been added in the table 2.

Point 4: Sensory properties are too slight explained. Please improve it

Response 4: We improved explanations for the changes in sensory properties and analyzed the relationship between them and other properties. The final statement is as follows: The scores of softness, stickiness, and elasticity of substitutions were slightly lower than those of the control group but no significant difference was found, which could be due to the significantly higher hardness, firmness value and lower extensibility. These results were comparable with the TPA analysis of cooked noodles. In addition, panelists reported a special flavor of noodles containing acorn flour, which may explain the lower flavor scores of acorn noodles. However, this flavor is not unacceptable according to their moderate scores. Furthermore, appearance and color were negatively influenced by the application of acorn flour due to the its darker color, which may be accountable for the lower overall liking scores of the substitutions than the control. Among all substitutions, the noodle with 10% acorn flour received the highest overall liking scores and scores of it were found statistically (p<0.05) similar to the control. Nevertheless, all noodles prepared with acorn flour were liked moderately and liked slightly by panelists.

Point 5: Why you used PCA? 5 samples are sufficient for PCA?

Response 5: PCA is a statistical method which try to recombine the original variables into a new group of independent comprehensive variables from which several less comprehensive variables can reflect the information of the original variables as much as possible. This multivariate statistical method is used to investigate the correlation between multiple variables, and to study how to reveal the internal structure of multiple variables through a few principal components. PCA was firstly introduced by K. Pearson to deal with non-random variables instead of samples. In this paper, we introduced PCA to analyze a total of 29 indicators to get a overview of relationships between the various properties of noodles with different substitution levels. There were similar uses in other literatures. For example, Kaur et al. used PCA to visualize the variation in the properties among four kinds of flours (DOI: 10.1007/s13197-010-0227-6) while Kaushal and Sharma (10.1080/10942912.2012.665405) carried PCA to visualize the variation in the properties among six different noodles. Therefore, it can be concluded that the numbers of our variables is sufficient to carry out PCA analysis.

Round 2

Reviewer 2 Report

Accept as it is